# Nanomechanical and Topochemical Changes in Elm Wood from Ancient Timber Constructions in Relation to Natural Aging

**DOI:** 10.3390/ma12050786

**Published:** 2019-03-07

**Authors:** Liuyang Han, Kun Wang, Weibin Wang, Juan Guo, Haibin Zhou

**Affiliations:** 1Research Institute of Wood Industry, Chinese Academy of Forestry, Haidian District, Beijing 100091, China; liuyang.han@empa.ch (L.H.); IAWA@caf.ac.cn (J.G.); 2College of Material Science and Technology, Beijing Forestry University, Haidian District, Beijing 100083, China; wangkun@bjfu.edu.cn; 3Shanxi Ancient Building Maintenance Quality Supervision Station, Taiyuan 030001, China; weibinwang3999@gmail.com

**Keywords:** timber buildings, aged wood, nanoindentation (NI), Raman spectra

## Abstract

Knowledge of properties of building materials affected by aging is of great importance to conserve cultural heritages or replace their biopolymer components. The objective of the study was to investigate the chemical characterization change in the biopolymer components and identify whether these changes are correlated with alterations in the nanomechanical properties of the wood cell wall bio-composites in relation to natural aging. The effects of natural aging on the elm (*Ulmus*) wood component (dated from 1642 to 1681) of Chenghuang Temple, an ancient timber construction in China were investigated to understand the chemical and mechanical changes in the wood cell wall. Especially, confocal Raman microscopy and nanoindentation (NI) were used to track changes in the chemical structure and nanomechanical properties. The results showed that the morphological, chemical and physical properties of cell walls changed with aging. After aging, the cell structure showed evidential alternations, and the wood components, especially hemicellulose and lignin, were degraded, leading to deterioration of mechanical properties of aged wood compared with normal wood. Morphology deterioration and micromechanical changes only occurred on the surface with the depth of about 3.6 mm of the aged element. This study would be helpful to provide practical guidance for protecting the apparent performance of ancient timber structures.

## 1. Introduction

Over centuries, wood, due to its specific properties, has been used to make varies objects including countless cultural heritages. A great deal of these old timber structure buildings are unique architectural and artistic objects, especially in Shanxi province, China. The Yingxian wooden pagoda, for example, is 67.31 m tall and was built in 1056 A.C. The wood elements from such buildings underlie nearly 1000 years. Wood has been found in countless cultural heritages owing to its good durability under suitable environmental conditions.

Some scholars have reported that the macrostructure of wood is extremely steady provided that it is not altered by microbial enzymatic decomposition [1]. Wood exposed to natural weathering is degraded due to fluctuations in temperature, humidity, and UV irradiation, thus, the wood surface finally disintegrates [2]. These changes could negatively affect the durability of wood even in the absence of microbial erosion. The wood in historical buildings, in panel paintings, and in sculptures often show tiny cracks under a microscope [3]. When the surface of wood was exposed to ultraviolet light, the first sign of deterioration was the increase of the sizes of pits in earlywood. That was followed by the occurrence of micro checks, which enlarged generally due to the shrinkage of cell walls. In the weathering process, the leaching and redistribution of water obviously promoted the expansion of the micro checks. The structural breakdown is a gradual, extremely slow process and eventually, there will be a deterioration of the middle lamella, the separation of wood fibers, and an increase of wood pores, such as pits.

As a cellular biomaterial, wood is composed of cellulose, hemicelluloses, and lignin, where the stiff cellulose fibrils are embedded in a soft matrix to support the stiffness of the cell wall, consisting of hemicelluloses and lignin [4]. The lignin within the cell wall matrix act together with glucomannan as a kind of “shock absorber” in the course of load transferring, not directly coupled to the main stress contributor cellulose. Mainly, lignin was employed as a viscoelastic filler within the supramolecular architecture by filling the spaces between cellulose chains in the cell wall [5]. Lignin, as an encrusting substance distributed in the matrix, greatly increases the rigidity of wood cell walls [6]. During aging wood is also subjected to oxidation, hydrolysis, depolymerization, and other chemical processes, which inevitably will take place with time [7]. Therefore, some changes must be expected even if a wood is exposed to a constant environment. The lignin content of samples stored under different conditions and with different wood species and ages ranging from 900 to 4400 years were investigated by IR spectroscopy, and it could be demonstrated that lignin in wood undergoes oxidative changes leading to a reduction in the amount of lignin separated by the hydrochloric acid process [8]. However, natural aging of fir wood showed a significant increase of extractives content and a decrease of hemicellulose (by a maximum of 24%), while the amounts of lignin and holocellulose were quite stable (decreasing only by 4%) [9]. Any breakdown of the cell wall may be finally caused by loss of chemical components. Wood may partially or completely disintegrate macroscopically if the matrix fails.

As is known, the unique architectural and artistic objects should not be protected on the premise of their destruction, so non-destructive or micro-destructive testing is suggested for aged wood studies. However, normal methods of obtaining wood properties, especially mechanical properties, are destructive and a large number of specimens are required for mechanical properties testing to reduce the variability of materials. Additionally, the minor reflection is sometimes difficult to monitor at the macroscopical level and it may be too late to take appropriate conservation steps for old wooden objects when the damage due to aging can be easily detected. It is important to note that all damage to wood’s molecular (chemical), anatomical, morphological, and geometric structure are reflected in its mechanical, physical, and durability properties. Thus, a clear understanding of wood aging on the micro and nanochemical and mechanical properties is necessary for the earliest conservation of wooden cultural heritage, as well as longer use of timber constructions.

The aim of the study was to investigate the chemical characterization change in the biopolymer components of the aged hardwood on a cell-structure level. A further objective was to determine whether these changes were related to alterations in nanomechanical properties of the wood cell wall bio-composites. The current study presents results from chemical and mechanical investigations on wood from deconstructed buildings of Shanxi province, China and, thus, will contribute to understanding the knowledge of wood aging.

## 2. Materials and Methods

### 2.1. Materials

The historic building of Chenghuang Temple is located in Jiexiu City, Shanxi province (111°44′10″ E–112°10′14″ E and 36°50′01″ N–37°11′04″ N), China (Figure 1A), which was first built in 1370 A.C. and was then repaired in 1495 A.C. and again at another later time. A purlin-type beam of these elements, which were replaced according to related standards in China during the last massive overhaul, was selected as the study beam (Figure 1B). One side of the beam was covered with visual or microscopically-determined fungal rot and insect galleries and another side showed visual aged characteristics. A sample block with a length of 30 cm was collected from the aged side without obvious attacks of fungi or insects. AMS 14C radiocarbon dating of the beam by a HVEE Xi’an AMS system (code Beta-215652) at the Xi’an AMS Center in China determined age of cal BP 269: cal BP 308 calibrated with CALIB 7.02 software using the IntCal13.14c dataset.

### 2.2. Method

#### 2.2.1. The Fabrication of Test Samples

Two wood discs were cut from the block, about 5 cm in thickness and about 30 cm in diameter. One of the discs was designated for the analysis of the transversal microscopical structure, chemical composition, and distribution (Figure 2A). Another disc was used to determine the ultrastructural, chemical, and mechanical properties of the aged wood (Figure 2B).

#### 2.2.2. Wood Identification and Transversal Microscopical Structure Analysis

Samples for wood identification and transversal microscopical structure were made from the purlin-type beam. For wood identification, about 20 mm thick cross-, radial-, and tangential sections were cut by hand using razor blades. Three sections (transverse, radial, and tangential sections, the thickness of 10–20 μm) were cut with the help of a Leica SM 2010R microtome (Buffalo Grove, IL, USA). They were observed under an Olympus BX50 microscope (Tokyo, Japan) after being stained with safranin. Terminologies in the IAWA Committee were used for anatomical descriptions [10]. Wood species was identified by comparison with wood specimens in the Wood Collection of Chinese Academy of Forestry [11]. As for the transversal microscopical structure, the samples were cut from #2, #3, #10, and #21, as shown in Figure 2A. The sample preparation method was similar to wood identification.

#### 2.2.3. Compositional Analysis

As shown in Figure 2A, the disc was cut into 22 slices and the slices were numbered #1 to #22 from the outer to the inner part of the wood. The structural carbohydrates and total lignin were determined by the National Renewable Energy Laboratory (NREL, Golden, CO, USA) protocol [12].

#### 2.2.4. Topochemical Distribution by Confocal Raman Microscopy

Three samples for local chemical distribution analysis in the transversal section were made as shown in Figure 2A (collected from #3, #10, and #21). Four samples for chemical distribution corresponding to mechanical property characteristics were prepared as shown in Figure 2B (named S1, S2, S3, and S4). For Raman imaging, 8 µm thickness cross-sections were cut from the wood samples using a sliding microtome (SM 2010 R, Leica, Germany) to obtain a full wafer. The cross-sections of wood samples were placed on glass slides covered with glass coverslips with a thickness of 0.17 mm. After being fully wetted with ultrapure water, the sample was sealed with nail polish as described in a previous study [13]. The Raman spectra were acquired with a Horiba Jobin Yvon confocal Raman microscope (Kyoto, Japan) equipped with a linear-polarized 532 nm laser. For mapping analysis, 0.5 μm steps were chosen and the final spectrum from each location was obtained by averaging two-second cycles. Raman images were created by a filter which can remove cosmic rays, and the specific areas in the wood spectra were compounded using the default Horiba software sum filters. Before further analysis, baseline correction of the calculated average spectra was analyzed using a seven-point lines-method, the Savitsky–Golay algorithm, as reported before [14,15]. The overview chemical images could be used to separate cell wall layers and, at the same time, label the distinct cell wall regions to calculate the average spectra from the specific areas.

#### 2.2.5. Scanning Electron Microscopy (SEM)

As shown in Figure 2B(b5), cross-sections of wood specimens were smoothed by hand with a razor, coated with platinum, mounted on aluminum stubs, and then observed by a Hitachi S4800 SEM (Tokyo, Japan) at a voltage of 7 kV.

#### 2.2.6. Nanoindentation (NI)

As shown in Figure 2B, the four latewood specimens (S1, S2, S3, and S4) for NI experiment were collected from one of the two logs. The four samples were cut to about 2 mm × 5 mm × 8 mm in radial, tangential, and longitudinal dimensions. A tilt apex was created positioning in the latewood by using a knife. The surface of the cross-section of the wood was smoothed with the help of a diamond knife under an ultramicrotome.

The NI test was carried out on a Hysitron TI 950 nanomechanical device (Bruker, Eden Prairie, MN, USA) equipped with a Berkovich-type diamond indenter tip whose radius is 100 nm. The NI samples were conditioned in the nanoindentator room for more than 24 h to minimize thermal shifts during the experiment [16]. Experiments were accomplished in the mode of the load-controlling method. Firstly, the maximum force of 200 μN was loaded in 5 s. Then the load was held for 2 s and unloaded in 5 s.

Figure 3 describes the procedure of the in-situ imaging NI experiment. The specimen surface was first observed by a light microscope (Figure 3A). Then, in the test mode of the scanning probe microscope, the accuracy of high indentation positioning was guaranteed. The marked regions were selected for the NI experiment and the locations to be indented were carefully selected, as shown in Figure 3B. At least four positions were required for each test region according to previous reports [17,18]. In each test area, as shown in Figure 3C, the space between every adjacent test point was at least 20 times the maximum indentation depth. Twenty to thirty indentations were made on a transverse section of 2–3 mature cell walls for each specimen. For reliable data analysis, results taken from outside of the secondary layer and cracks were eliminated, and the average value of all the validated results was used. The most extreme values in the data set (maximum and minimum values), the lower and upper quartiles, and the median were analyzed by using Minitab statistical software (Minitab 18.1, State College, PA, USA). The values of elastic modulus (*E*) and hardness (*H*) were obtained according to the reported method [19]. *H* can be obtained as follows:(1)$$H=\frac{P}{A}$$
where *P*_max_ is the peak load and *A* is the corresponding contact area. The reduced indentation modulus (*E*_r_) can be determined [20]:(2)$$E_{r}=\frac{dP}{dh}\times \frac{1}{2}\times \frac{\sqrt{\pi }}{\sqrt{A}}$$
where $\frac{dP}{dh}$ is the slope of the tangent of the unloading curve in the load-displacement diagram.

*E* was then calculated by:(3)$$\frac{1}{E_{r}}=\frac{{1-v}^{2}}{E}+\frac{{1-v}_{i}^{2}}{E_{i}}$$
where *E*_i_ is the elastic modulus and *v*_i_ is Poisson’s ratio of the nanoindentation tips. For diamond tips, *E*_i_ is 1141 GPa and the value of *v*_i_ is 0.07. E_s_ and *v*_s_ are the elastic modulus and Poisson’s ratio of the specimens. The value of *v*_s_ was assumed as 0.25 for the wooden materials according to former researches [20,21,22,23].

## 3. Results and Discussion

### 3.1. Wood Identification

The wood blocks from the beam member were identified microscopically to *Ulmus* sp. (elm wood). Three sections of elm wood were shown in Figure 4. Wood ring-porous; round or oval vessels solitary in earlywood, containing tyloses; and vessels mostly clusters in latewood, wavy tangential pattern. Perforation plates simple. Intervessel pits alternate. Vascular tracheids are similar to narrower vessel elements in shape. Helical thickenings are only in narrower vessel elements and vascular tracheids. Vessel-ray pits are similar to intervessel pits [10]. Axial parenchymas are mostly vasicentric, occasionally diffuse-in-aggregates and diffuse. Rays are non-storied, 1–6 cells wide, up to 70 cells high. All ray cells are procumbent.

### 3.2. The Effect of Aging on Micro-/Ultra-Scope Structure Properties

As demonstrated in Figure 5, diffuse-porous wood with obvious growth ring boundaries in cross-section was observed. Vessels arranged in a branching pattern, forming distinct tracts, separated by areas devoid of vessels, which are circular to angular in the transactional outline. From pith to the cambium, the diameter of the vessel decreased, accompanied by a gradual increase in the number of vessel. Moreover, xylem rays are multi-seriate. Parenchyma cells associated with or around the vessels were occasionally discovered. The microstructure of specimens from #2 to #21 presented no obvious change, which means that there was no effect on the microstructure of deep layers when exposed to weather.

SEM images of aged (S1) and sound elm wood specimens revealed significant structural differences between them (Figure 6). It was observed that the sound elm wood showed smooth, highly ordered, and compact morphology (Figure 6B–D), resembling the structure of contemporary wood, whereas cell walls looked fragile in the cross-section of aged wood (Figure 6A) and cracks exhibited among cell walls (Figure 6(a2)). Although S1 and S2 were sectioned from the same growth ring, they had different morphology. The discrepancies in surface structures (S1) may be explained by the degradation of cellulosic and hemicellulose particles during the aging process. These results were similar to a previous study on archaeological wood where surface portions of the wood demonstrated the morphological alteration of some cells while parts of wood from below the surface layer had no cell wall degradation [24].

### 3.3. The Effect of Aging on Chemical Composition and Distribution

Table 1 gives the chemical compositions of wood slices (#1–#22) from outer to the inner locations, displaying the chemical differences at the tissue level. Clearly, glucan was the dominant component, followed by lignin and xylan. The relative content of glucose and xylose was slightly decreased from the inside (#22) to the outside (#1). Correspondingly, the relative content of lignin gradually increased.

For the overall measurement of the cell-wall structure, the Raman images were obtained by integrating the CH and CH_2_ vibrations from 2920 to 2768 cm^−1^. Correspondingly, the types and morphologically distinct regions of various cell wall were clearly visualized. High carbohydrate concentration, ascribing to the high intensity, was observed especially in the secondary cell wall (S-O: Outer-layer; S-M: Middle-layer; S-I: Inner-layer), and following by the sub-cellular level including the cell corner middle lamella (CCML) and the compound middle lamella (CML) (Figure 7A,C,E,G,I). Meanwhile, the Raman image after being integrated into with the range of 1660 to 1540 cm^−1^ exhibited the heterogeneity in lignin distribution (Figure 7B,D,F,H,J), with a higher intensity in the CCML, CML, and S-I. From inner to outer, the Raman images clearly displayed an increase both in the concentration of carbohydrates and lignin (Figure 7J–E, from #21 to #3), which was probably caused by natural growth of trees. However, there was a consistently decrease from #3 to #1 for carbohydrates and lignin, which was probably due to the long-term degradation of the cell wall. This is in agreement with the results from previous research that lignin is a sensitive wood component to UV light when outdoors and exposed to direct sunlight [25]. Since the correlation between lignin and carbohydrates concentration at cellular level could provide more detailed information, the Raman images was further constructed according to lignin/carbohydrates ratio and a relative increment was found in CCML and CML probably indicated that the hemicellulose polymer was more vulnerable to be degraded than lignin during the aging process, due to its random and amorphous structure.

### 3.4. The Effect of Aging on Micromechanical Properties

NI technique can be used to obtain the longitudinal elastic modulus and hardness of the cell wall at a submicrometer level [18]. Furthermore, an atomic force microscope equipped with a nanoindenter was applied in order to ensure the location and quality of dents made in the cell walls. E values and H values for the secondary cell wall (Figure 8) both showed an increasing trend from the inner to the outer parts of the wood disc, but a significant reduction was shown from S2 to S1. The mean indentation modulus and hardness values of S1, which occasionally occurred in the outermost edge of the outer layer of a purlin-type beam, were 18.76 GPa with a coefficient of variation (COV) of 4.58% and 0.47 GPa with a COV of 5.20%, respectively. S2 (normal wood), which was collected from the same growth ring with S1(aged wood) but was protected by the first growth ring (as shown in Figure 8), had the highest indentation modulus of 19.8 GPa with a COV of 6.02% and the highest hardness of 0.5 GPa with a COV of 4.67% for the secondary cell walls. The average growth ring width of the outer layer of our purlin-type beam was about 3.6 mm, and the mechanical properties of S2, which was protected by the outer layer of the wood without aging, were better than the aged S1, thus, we presumed that the aging depth was at least 3.6 mm in our aged wooden specimen. Many factors, especially density, chemical composition, the moisture content, and crystallinity, could have a significant influence on the mechanical properties of wood cells [22,26,27]. The strength behavior of the cell wall probably reduces because of both the crystallinity of cellulose in the fibrils and the cross-linking of lignin in the matrix [28], and this may explain the result of the lower values of S1 than S2. 

Raman microspectroscopy is normally employed to provide the spatical distribution of carbohydrates and lignin in their native form, which could be simultaneously mapped by selecting the specific Raman bands of these components. Integrating over the characteristic band range from 1720 to 1540 cm^−1^ (aromatic ring stretching) highlighted the higher lignin concentration in CCML and CML. It is clear that the preferential degradation of lignin by aging in fiber wall reduced the concentration of lignin in CCML and CML layers from S2 to S1 specimen, further confirming the reduced indentation modulus from the NI analysis. This agreed with the results of the previous study that increasing lignin content might contribute to the higher measured hardness [20,22,29].

The strength and rigidity of the cell wall are probably related to the cross-linking of lignin and the cell wall density [30]. Degradation of aged wood cell wall components can result in the dissolution of materials leaving the cell wall that are more porous, similar to normal pulping. The reduction of carbohydrates and the concentration of lignin in the fiber wall after aging may lead to a decrease of density, less support for the aggregates of crystalline cellulose, and lower rigidity and elastic modulus of the cell wall, which finally results in the morphology deterioration of wood near the surface of the architectural element.

## 4. Conclusions

The wood blocks from the old timber building were identified as *Ulmus* sp. (elm wood) with an age dating from 1642 to 1681. The mechanical and chemical properties of cell walls of aged wood are challenging to be characterized using customary methods due to the difficulty to get the aged wooden components from an ancient temple and the rare chance to obtain the specimens in the same growth ring with different aging conditions. However, nanoindentation and Raman imaging could be used to analyze the above properties at the cell wall level in a non-destructive or micro-destructive way. The micromechanical and microchemical property changes in wood cell wall of aged wood specimens chosen from an important representative load-bearing frame of a typical historical building, a wooden purlin-type beam of a Chinese temple, were explored. The surface of this archaeological elm wood had been significantly aged or weathered. The morphology deterioration of wood near the surface of the aged element occurred. Moreover, carbohydrates and lignin of the outer layer of the purlin-type beam had obviously decreased. Furthermore, the values of longitudinal hardness and elasticity in the surface of the aged wooden specimen (0.47 and 18.76 GPa, respectively) were significantly lower than those of the unexposed position in the same growth ring (0.5 and 19.8 GPa, respectively). Additionally, microchemical property changes only occurred near the shallow surface layer of the beam which was consistent with the changes of nanomechanical property. Although the aging action may not directly affect the structural safety of ancient buildings, it might cause other problems, such as the fallout of murals and paintings, the generation and aggravation of biodegradation, etc. Thus, in the process of protecting ancient buildings, it is necessary to carry out targeted protection treatment on aged elements.

## Figures and Tables

**Figure 1 materials-12-00786-f001:**
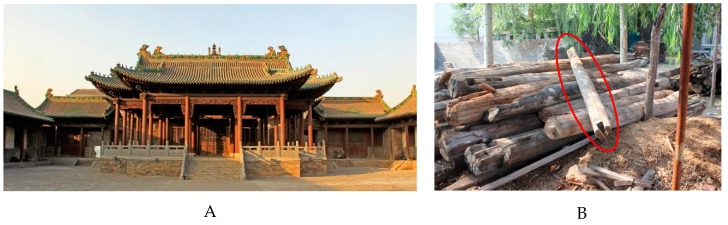
Chenghuang Temple (**A**) and the purlin-type beam (**B**) for this research. The beam in the red oval (**B**) was replaced from the old temple.

**Figure 2 materials-12-00786-f002:**
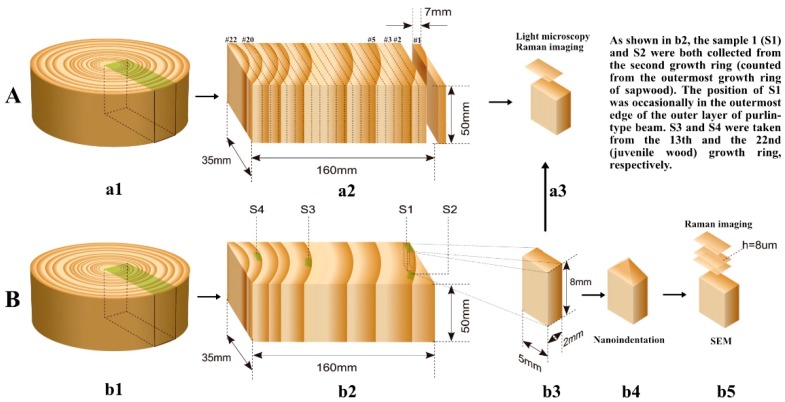
Scheme of preparation of wood samples for measurements. The pictures in row (**A**) indicate the preparation of samples for measurements utilizing microscopical structure analysis, compositional analysis (Numbered #1 to #22 from the outer to the inner part of the wood disc), topochemical distribution by confocal Raman microscopy. The pictures in row (**B**) illustrate the preparation of samples for measurements applying NI, Raman imaging, and SEM.

**Figure 3 materials-12-00786-f003:**
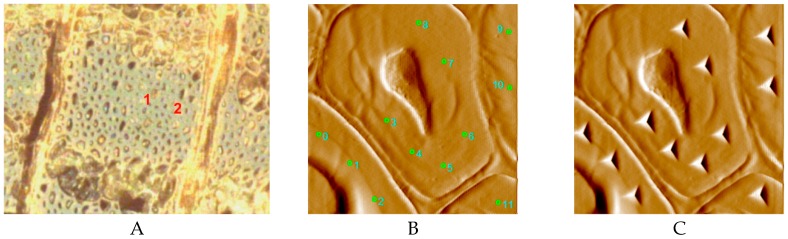
Procedure of an in-situ imaging NI experiment on wood cell. (**A**) region 1 and region 2 are the selected regions for NI test; (**B**) locations for NI were selected and marked; (**C**) the indentation of a region.

**Figure 4 materials-12-00786-f004:**
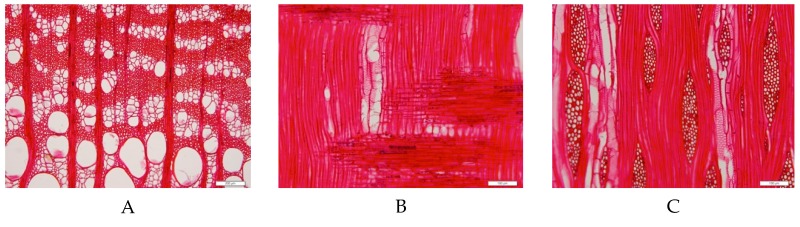
Microstructure of elm wood. (**A**) Transverse section; (**B**) radial section; and (**C**) tangential section.

**Figure 5 materials-12-00786-f005:**
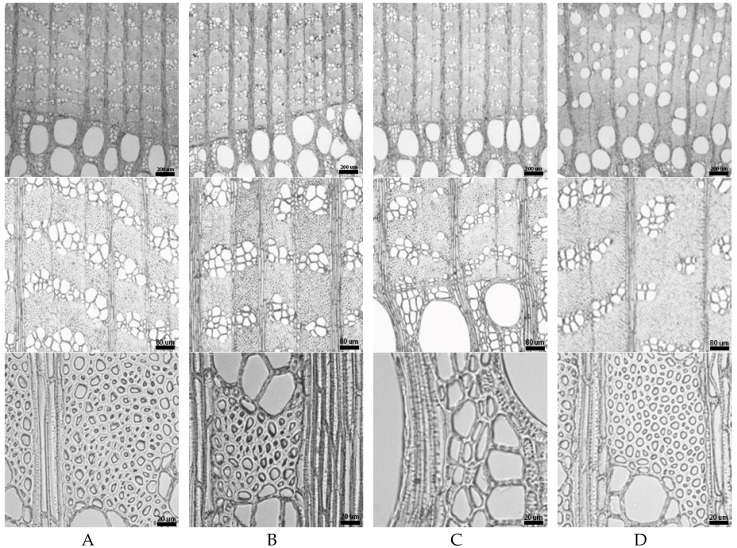
Distinct growth ring boundaries in the transverse section of elm wood. Pictures in column (**A**) #2, (**B**) #3, (**C**) #10, and (**D**) #21 were photographed from the 2nd, the 3rd, the 10th, and the 21st sample splints, respectively.

**Figure 6 materials-12-00786-f006:**
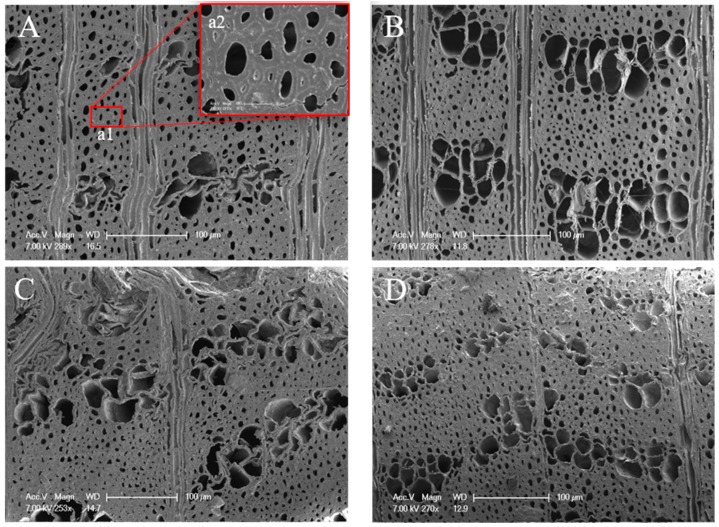
SEM pictures of the transverse section of elm wood samples. (**A**–**D**) were the results from S1, S2, S3 and S4 respectively. a2, which was randomly choose in (**A**), is 8 times larger than a1.

**Figure 7 materials-12-00786-f007:**
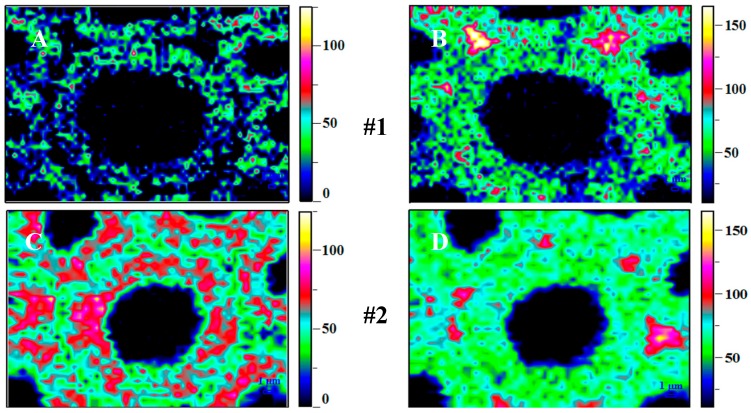
Raman images calculated by integrating from 2920 to 2768 cm^−1^ (**A**,**C**,**E**,**G**,**I** carbohydrates) and from 1660 to 1540 cm^−1^ (**B**,**D**,**F**,**H**,**J** lignin). (**A**,**B**) in Row #1 are the Raman imaging for the 1st sample splint showed in Figure 3A, a2. (**C**,**D)** (in Row #2), (**E**,**F)** (in Row #3), (**G**,**H)** (in Row #10), and (**I**,**J)** (in Row #21) are the Raman imaging for the 2nd, 3rd, 10th, and 21st sample splints. respectively.

**Figure 8 materials-12-00786-f008:**
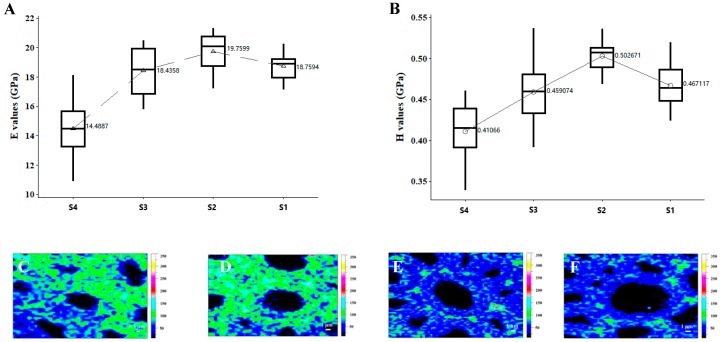
(**A**) Indentation modulus (*E*) values, determined by NI, of the secondary cell wall of elm wood. (**B**) Hardness (*H*) values, determined by NI, of the secondary cell wall of elm wood. Raman images calculated by integrating from 1720 to 1540 cm^−1^ (lignin): (**C**) S4; (**D**) S3; (**E**) S2; (**F**) S1.

**Table 1 materials-12-00786-t001:** Chemical compositions of all specimens.

Num.	Ara	Gal	Glc	Xyl	Man	ASL	AIL
1	0.0%	0.6%	40.3%	13.9%	2.3%	8.5%	34.5%
2	0.0%	0.5%	38.5%	14.6%	1.9%	5.6%	38.9%
3	0.3%	0.8%	41.7%	18.2%	2.9%	1.6%	34.5%
4	0.3%	0.8%	43.3%	18.7%	3.1%	1.3%	32.5%
5	0.4%	0.9%	42.3%	19.6%	2.9%	1.7%	32.2%
6	0.6%	0.9%	42.4%	19.8%	2.6%	1.8%	32.1%
7	0.4%	0.8%	42.3%	20.2%	3.1%	1.5%	31.7%
8	0.5%	0.7%	43.7%	18.6%	3.7%	1.6%	31.2%
9	0.3%	0.7%	44.8%	17.3%	3.8%	1.6%	31.4%
10	0.3%	0.8%	44.3%	17.8%	3.5%	1.8%	31.5%
11	0.3%	0.7%	45.3%	18.7%	3.6%	1.6%	29.8%
12	0.3%	0.6%	45.8%	18.1%	3.4%	1.6%	30.2%
13	0.3%	0.8%	44.7%	17.7%	3.6%	1.7%	31.2%
14	0.3%	0.8%	46.3%	17.7%	3.3%	1.6%	29.9%
15	0.3%	0.9%	46.4%	17.5%	2.7%	1.6%	30.5%
16	0.4%	0.9%	46.7%	17.6%	2.7%	1.6%	30.1%
17	0.3%	0.7%	47.0%	18.4%	3.1%	1.4%	29.0%
18	0.3%	0.8%	46.6%	18.1%	2.8%	1.7%	29.8%
19	0.3%	0.8%	46.5%	18.8%	2.8%	1.6%	29.3%
20	0.3%	0.9%	47.3%	18.7%	2.4%	1.7%	28.8%
21	0.4%	1.3%	43.6%	19.4%	2.4%	2.2%	30.8%
22	0.3%	0.8%	43.7%	17.0%	1.8%	2.1%	34.3%

Note: Ara, Arabian; Gal, Galactan; Glc, Glucan; Xyl, Xylan; Man, mannose; ASL, Acid-soluble lignin; AIL, Acid-insoluble lignin.

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
