# Peer review of "Nanomechanical and Topochemical Changes in Elm Wood from Ancient Timber Constructions in Relation to Natural Aging"

_materials, 2019, doi:10.3390/ma12050786_

Reviewer 1 Report

The paper is well prepared, but requires some editoral chages.

Subheadings in sections 2.2.2.-2.2.6 and 3.1. have not been defined. Still “subsubsection” is standing.

Grammar and style of the manuscript should be revised. Some expressions and phrases are not clear or erroneous. To name a few:

Avoid short forms – e.g. “shouldn’t”, “what’s”. Use “should not” and “what is” instead throughout the text.

l. 18: cal cal BP 269: cal BP 308 – not clear meaning of BP What does it stand for? Before Christ? Extend that.

l. 33-34: Over centuries, due to its specific properties, wood has been used for production of varies

objects, which are part of the cultural heritage nowadays.

l. 39-40: It has previously been shown that the macroscopic structure of wood remarkably stable as long as it is not changed by enzymatic breakdown due to microorganisms.

l. 44: often shows fine cracks

l. 66: It could be shown that lignin in wood undergoes….

l. 146-148: The overview chemical images separated cell wall layers and marked the defined distinct cell wall areas to calculate the average spectra from the areas of interest.

l. 261: heimcellulosic

l. 267: microscope attached by the nanoindenter. “Attateched to” or “equipped with”?

I suggest accept when these changes are made.

Author Response

Point 1: 

Subheadings in sections 2.2.2.-2.2.6 and 3.1. have not been defined. Still “subsubsection” is standing.

Grammar and style of the manuscript should be revised. Some expressions and phrases are not clear or erroneous. To name a few:

Avoid short forms – e.g. “shouldn’t”, “what’s”. Use “should not” and “what is” instead throughout the text.. 

Response 1: Thank you for your professional suggestions and we have revised all of them in our manuscript.

Point 2:l. 18: cal cal BP 269: cal BP 308 – not clear meaning of BP What does it stand for? Before Christ? Extend that..

Response 2:Thanks a lot and we have changed the radiocarbon dating cal BP into a more understandable expression.

 Point 3:

l. 33-34: Over centuries, due to its specific properties, wood has been used for production of variesobjects, which are part of the cultural heritage nowadays.

l. 39-40: It has previously been shown that the macroscopic structure of wood remarkably stable as long as it is not changed by enzymatic breakdown due to microorganisms.

l. 44: often shows fine cracks

l. 66:It could be shown that lignin in wood undergoes….

l. 146-148: The overview chemical images separated cell wall layers and marked the defined distinct cell wall areas to calculate the average spectra from the areas of interest.

l. 261: heimcellulosic

l. 267: microscope attached by the nanoindenter. “Attateched to” or “equipped with”?

Response 3:Thank you for your professional suggestions and we have revised all the parts according to your suggestions in our manuscript.

Reviewer 2 Report

The authors focus on interesting problematics, namely nanomechanical and topochemical changes in old building timber from elm wood. The results obtained are, in any case, beneficial from the point of view of the deepening of the background to the possible targeted protection of wooden historic buildings.

I would like to make few corrections and additions.

Formal shortcomings:

- missing names of subchapters, such as 2.2.2, 2.2.3, ..., 3.1,

- irrelevant sentences: "In 2015, the building was replaced from the old temple. began a massive overhaul for good protection. "(line 95-96),

- occasional not quite ideal expression, for example, the first sentence of the abstract: "Properties of building materials affected by aging are of great importance to conserve cultural heritages or replace their biopolymer components." (line 13-14). If so, "Knowledge of properties...".

Missing details (facts):

- the statistical significance of the results obtained,

- comparison of the obtained results with "normal" present elm-wood, i.e. the relevant decreases in the parameters examined and the properties caused by natural aging, minimally the representation of hemicelluloses (xylanes and mannanes), i.e. "filling" and skeletal types, it would be interesting to compare, as well as elasticity and hardness of the cell wall,

- the analysis of density and dendrometric quantities on larger dimensional specimens across cross-section of the stem, even though I am aware of the concrete specific content of this manuscript, i.e. a research of wood a few structure levels below (nanoindentation and Raman spectroscopy),

- some concrete results of the research to include into the conclusions and apparently to the abstract in numerical terms (values),

- closer specification of deterioration of structure and properties of wood, depth specification - the formulation "…occurred near the surface of the aged element“ is not enough".

The methods of research used, and the introduction to the problem, I consider as appropriate and sufficient. I welcome the use of modern research technologies and relevant discussion on the results. However, I would add to the results the statistical significance and to the discussion at least comparison with the available literature (comparison with present elm-wood). If the number of samples within a given series (properties) is less than 30, it is necessary to verify the statistical significance of the obtained results. For example, through a quantile of Student's distribution (e.g. significant level 0.95), related variability of measurement (COV) and selected accuracy of measurement (e.g. 5%).

However, the reproducibility of the results will be difficult to estimate, particularly because of the high variability in wood properties, aging effects, specific climatic conditions, etc. It will therefore always be tied to the specific conditions of using, like in this case.

Author Response

Point 1: 

Formal shortcomings:

- missing names of subchapters, such as 2.2.2, 2.2.3, ..., 3.1,

- irrelevant sentences: "In 2015, the building was replaced from the old temple. began a massive overhaul for good protection. "(line 95-96),

- occasional not quite ideal expression, for example, the first sentence of the abstract: "Properties of building materialsaffected by aging are of great importance to conserve cultural heritages or replace their biopolymer components." (line 13-14). If so, "Knowledge of properties...".. 

 Response 1: 

Thank you for your professional suggestions. We have revised our manuscript according to your suggestions of formal shortcomings.

 Point 2:

Formal shortcomings:

- missing names of subchapters, such as 2.2.2, 2.2.3, ..., 3.1,

- irrelevant sentences: "In 2015, the building was replaced from the old temple. began a massive overhaul for good protection. "(line 95-96),

- occasional not quite ideal expression, for example, the first sentence of the abstract: "Properties of building materialsaffected by aging are of great importance to conserve cultural heritages or replace their biopolymer components." (line 13-14). If so, "Knowledge of properties...".. 

Response 1: 

Thank you for your professional suggestions. We have revised our manuscript according to your suggestions of formal shortcomings.

Point 2:

Missing details (facts):

2.1- the statistical significance of the results obtained,

2.2- comparison of the obtained results with "normal" present elm-wood, i.e. the relevant decreases in the parameters examined and the properties caused by natural aging, minimally the representation of hemicelluloses (xylanes and mannanes), i.e. "filling" and skeletal types, it would be interesting to compare, as well as elasticity and hardness of the cell wall,

2.3- the analysis of density and dendrometric quantities on larger dimensional specimens across cross-section of the stem, even though I am aware of the concrete specific content of this manuscript, i.e. a research of wood a few structure levels below (nanoindentation and Raman spectroscopy),

2.4- some concrete results of the research to include into the conclusions and apparently to the abstract in numerical terms (values),

2.5- closer specificationof deterioration of structure and properties of wood, depth specification. the formulation "…occurred near the surface of the aged element“ is not enough".

2.6 The methods of research used, and the introduction to the problem, I consider as appropriate and sufficient. I welcome the use of modern research technologies and relevant discussion on the results. However, I would add to the results the statistical significance and to the discussion at least comparison with the available literature (comparison with present elm-wood). If the number of samples within a given series (properties) is less than 30, it is necessary to verify the statistical significance of the obtained results. For example, through a quantile of Student's distribution (e.g. significant level 0.95), related variability of measurement (COV) and selected accuracy of measurement (e.g. 5%).

Response 2:

Thanks a lot for your professional suggestions and questions. Our answers are listed below.

-2.1, 2.6 (the statistical significance of the results): 

We have the statistical significance result, but we didn’t illustrate it specifically because we explained all the parameters, such as “minimum”, first quartile (Q1), median, mean value, third quartile (Q3), and “maximum” in the boxplots. The important COV values have been added in line 280 and line 283 according to your kind suggestion.

-2.2, 2.3, 2.6 (Comparison of the obtained results with normal wood and add the analysis of density and dendrometric quantities on larger dimensional specimens across cross-section of the stem): 

Thanks a lot for your kind suggestions! To be honest, we didn’t think it is meaningful to compare the results of aged wood with normal wood since the high variability between different wood and even different height of the same trunk, therefore, we designed this experiment to compare properties of exposed and non-exposed parts of the same ring in the same wood and the same height. Besides, there was no available literature to compare with (at least we don’t find yet) since no research on elm using NI or AFM without imbedding medium. The analysis of density and dendrometric quantities on larger dimensional specimens across cross-section of the stem is a very interesting point and we could combine the idea with researching on the macramechnical properties using dynamic mechanical analysis in future! Thanks a lot for this meaningful suggestion!

-2.4 (add values to abstract and conclusions):

Thank you very much for your kind suggestion and we added these parts in our manuscript according to your kind reminding.

 -2.5 (closer specification):

Thank you for your suggestions and we measured the growth ring width of the related wood samples and added the result in our manuscript in line 29 and line 286-290.
